# OmniQuality-R: Advancing Reward Models through All-Encompassing Quality Assessment

## Abstract

Current visual evaluation approaches are typically constrained to a single task — focusing either on technical quality for low-level distortions, aesthetic quality for subjective visual appeal, or text-image alignment for semantic consistency. With the growing role of reward models in guiding generative systems, there is a need to extend into an all-encompassing quality assessment form that integrates multiple tasks. To address this, we propose OmniQuality-R, a unified reward modeling framework that transforms multi-task quality reasoning into continuous and interpretable reward signals for policy optimization. Inspired by subjective experiments, where participants are given task-specific instructions outlining distinct assessment principles prior to evaluation, we propose OmniQuality-R, a structured reward modeling framework that transforms multi-dimensional reasoning into continuous and interpretable reward signals. To enable this, we construct a reasoning-enhanced reward modeling dataset by sampling informative plan-reason trajectories via rejection sampling, forming a reliable chain-of-thought (CoT) dataset for supervised fine-tuning (SFT). Building on this, we apply Group Relative Policy Optimization (GRPO) for post-training, using a Gaussian-based reward to support continuous score prediction. To further stabilize the training and improve downstream generalization, we incorporate standard deviation (STD) filtering and entropy gating mechanisms during reinforcement learning. These techniques suppress unstable updates and reduce variance in policy optimization. We evaluate OmniQuality-R on three key IQA tasks: aesthetic quality assessment, technical quality evaluation, and text-image alignment. Experiments show OmniQuality-R improves robustness, explainability, and generalization, and can guide text-to-image generation models at test time without retraining by serving as an interpretable reward function.

## 1 Introduction

The rapid growth of visual data, including user-generated content (UGC) and AI-generated content (AIGC), presents new challenges for image quality assessment (IQA) across diverse domains and tasks (Agnolucci et al., 2024; Min et al., 2024; Chen et al., 2024a; Sun et al., 2024; Peng et al., 2024; Li et al., 2024b; Yuan et al., 2024; Yu et al., 2024; Fang et al., 2024). Traditional IQA approaches, which rely on hand-crafted features (Mittal et al., 2012b;a) or neural networks (Talebi & Milanfar, 2018; Su et al., 2020; Network, 2022; Ke et al., 2021; Wang et al., 2023; Yang et al., 2022; Zhong et al., 2025; 2024) trained on synthetic or authentically distorted image datasets, often exhibit limited generalization to new data. Moreover, these approaches typically output a single scalar score, offering little insight into the underlying reasons for the quality judgment and lacking interpretability. As IQA increasingly plays a critical role in guiding image post-processing (e.g., super-resolution, restoration) (Jiang et al., 2025; Li et al., 2025a; Chen et al., 2024c), image compression (Wang et al., 2025b; Li et al., 2024c), and serving as a reward model for preference optimization in text-to-image (T2I) generation (Luo et al., 2025; Gu et al., 2024; Wang et al., 2025d; Xu et al., 2024; Ma et al., 2025; Chen et al., 2024d; Xie et al., 2025a;b; Qin et al., 2025), the demand for robust and explainable reward models through all-encompassing quality assessment has become more pressing. These conventional methods are typically constrained by the domain-specific characteristics of their training data, limiting their applicability to the diverse, high-variance visual content encountered in modern applications.

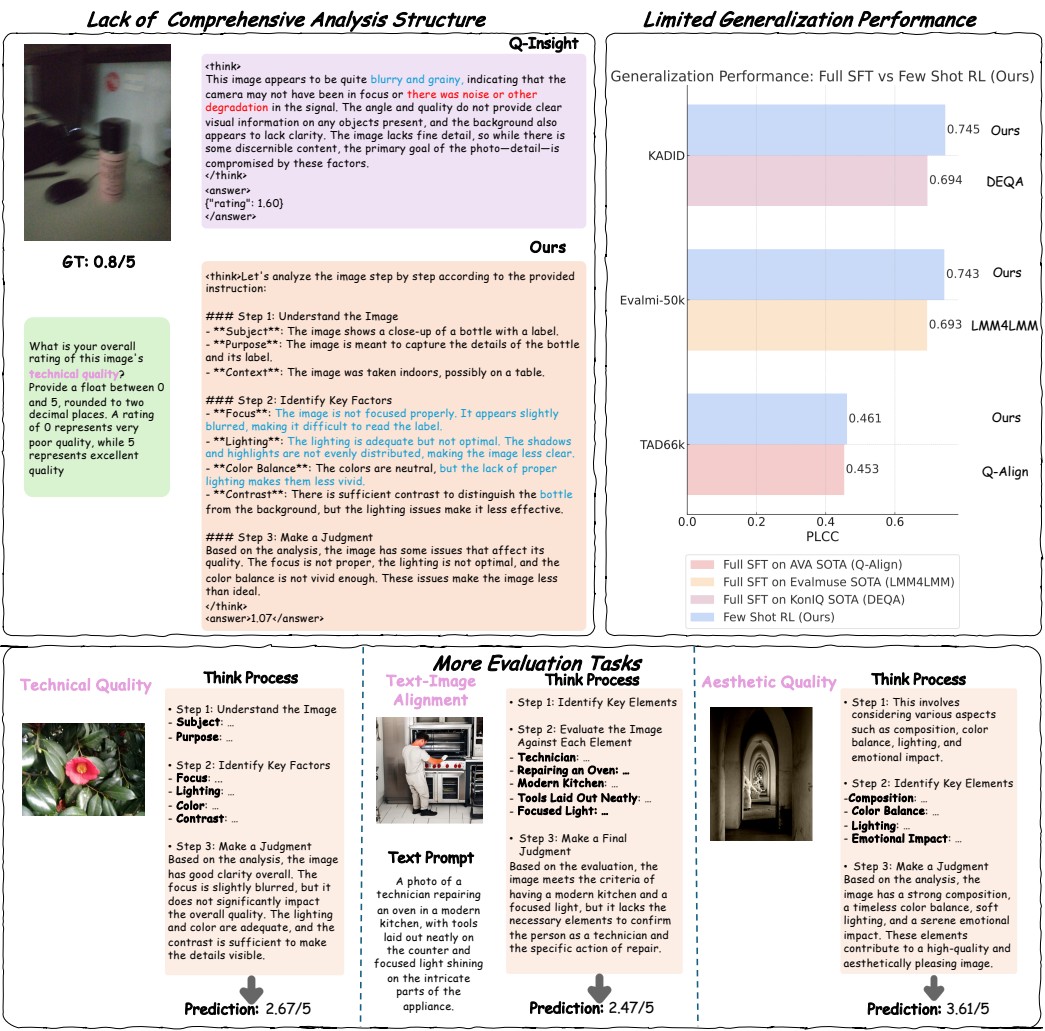

Figure 1: Overview of the challenges in multimodal image quality assessment and the proposed OmniQuality-R framework. **Left**: Existing methods lack a structured reasoning process, often yielding incomplete analysis, whereas OmniQuality-R introduces a step-by-step, interpretable reasoning structure. **Right**: OmniQuality-R achieves improved generalization performance, particularly with few-shot data through reinforcement learning (RL), outperforming existing SOTA models across multiple benchmarks. **Bottom**: OmniQuality-R supports multi-dimensional evaluation—technical quality, aesthetic appeal, and text-image alignment—each guided by a transparent think process and yielding final quality predictions.

The recent development of vision-language models, such as CLIP (Sun et al., 2023; Radford et al., 2021) and BLIP (Li et al., 2022; 2023), offers new opportunities for IQA. By leveraging rich vision-language pretraining, these models enable more robust quality evaluation across a wide range of content types (Wang et al., 2023; Zhang et al., 2023). Recent advances in quality-aware multimodal large language models (MLLMs) (Wu et al., 2024b; Zhang et al., 2024b; Chen et al., 2024b; Wu et al., 2024a; Jia et al., 2024; 2025; Wu et al., 2023a) have further expanded the landscape of IQA (Zhang et al., 2025). Several approaches explicitly design prompts or conduct supervised fine-tuning (SFT) to inject quality sensitivity into MLLMs. For instance, Q-Instruct (Wu et al., 2024b) and Co-Instruct (Wu et al., 2024c) adopt instruction tuning on diverse quality-related question answering and caption tasks, while Q-Align (Wu et al., 2023a), Q-Boost (Zhang et al., 2024a) and DeQA (You et al., 2025) focus on quality scoring tasks. Other works such as DepictQA (You et al., 2024b) and DepictQA-v2 (You et al., 2024a) introduce text reasoning ability for full-reference image quality assessment, and Q-Ground (Chen et al., 2024b) leverages quality-grounded visual grounding to enhance localization of quality issues. Despite these efforts, current quality-aware MLLMs typically

follow fixed reasoning patterns dictated by curated datasets and instructions, limiting their flexibility and depth in complex assessments. Even cutting-edge models like Qwen2-VL (Wang et al., 2024) and Qwen2.5-VL (Bai et al., 2025), while demonstrating strong general perception capabilities, still face limitations in low-level quality perception due to their unified-domain training. As shown in Fig. 1, existing multimodal reasoning models for explainable image quality assessment suffer from three major limitations: **incomplete dimension analysis**, **limited generalization**, and **narrow task scope**. Incomplete dimension analysis leads to the omission of key quality factors, reducing the interpretability and thoroughness of the evaluation. Limited generalization weakens the model's robustness across varying image types and distortion scenarios, undermining its applicability in real-world settings. Narrow task scope restricts the model to a limited range of IQA scenarios, preventing it from capturing the diverse requirements of multi-domain applications.

To address these challenges, we introduce OmniQuality-R, a reward model for all-encompassing image quality assessment that enhances multimodal reasoning through structured and interpretable evaluation across three core tasks: technical quality for low-level distortion, text–image alignment for semantic consistency, and aesthetic quality subjective visual appeal, as illustrated in Fig. 1. Moreover, it can improve generalization by supporting diverse quality-related tasks under a shared reasoning prototype/structure. To enhance the quality-aware reasoning capabilities of Multimodal Large Language Models (MLLMs) for quality assessment across diverse tasks and scenarios, we design a two-stage training process for OmniQuality-R. The overall training process includes a cold-start rejective sampling fine-tuning stage to teach implicit question analysis and explicit reasoning structures based on the analysis plan, followed by a unified reinforcement tuning stage with Group Relative Policy Optimization (GRPO) to explore reasoning pathways for score prediction. In the subsequent reinforcement tuning stage, we first introduce a Gaussian-shaped reward function to better reflect the continuous nature of score regression, enabling smoother and more informative policy optimization. However, as training progresses, the model tends to produce increasingly uniform score predictions, which leads to a sharp drop in entropy and vanishing advantage signals—both of which hinder effective learning. To address this, we incorporate entropy masking to encourage exploration of diverse chain-of-thought pathways and prevent premature policy convergence. Additionally, we apply STD-guided filtering to identify and filter out samples where predicted advantages collapse to near-zero, thereby focusing updates on more informative examples. Together, these mechanisms ensure stable training dynamics and further improve performance, especially on challenging out-of-distribution benchmarks. In summary, the contributions of this paper are summarized as follows.

- **OmniQuality-R Framework for Structured Evaluation:** We introduce the OmniQuality-R framework, which separates the planning and reasoning stages to provide structured, comprehensive, and interpretable evaluations across technical, aesthetic, and alignment dimensions. This separation promotes effective quality assessment in multimodal tasks.
- **Two-Stage Training for Enhanced Reasoning:** We design a novel two-stage training process, combining cold-start rejective sampling fine-tuning, and reinforcement learning with Group Relative Policy Optimization (GRPO). This process encourages the model to learn implicit question analysis and explicit reasoning structures, boosting performance on score prediction tasks.
- **STD-filtering and Entropy Gating for Stable Training:** We incorporate standard deviation (STD) sampling and entropy gating mechanisms during reinforcement tuning to stabilize training and enhance model's generalization performance on downstream tasks, particularly for continuous score regression tasks.
- **Versatile Assessment Across-Domain and Test-Time Guidance:** OmniQuality-R achieves the optimal performance on multiple quality assessment tasks across diverse domains. Furthermore, it can be used as a test-time reasoning module to guide and enhance text-to-image (T2I) generation models, improving alignment and compositional fidelity without additional training.

## 2 RELATED WORK

### 2.1 MLLMs

Recent progress in multimodal large language models (MLLMs) (Wang et al., 2024; Bai et al., 2025; Team et al., 2025; Zhu et al., 2025) has advanced along two major directions. The first focuses on developing general-purpose, state-of-the-art MLLMs such as Qwen2-VL (Wang et al., 2024), Qwen2.5-VL (Bai et al., 2025), InternVL3 (Zhu et al., 2025), Pixtral (Agrawal et al., 2024), and Kimi-VL (Team et al., 2025). These models are pre-trained on large-scale multimodal corpora and further

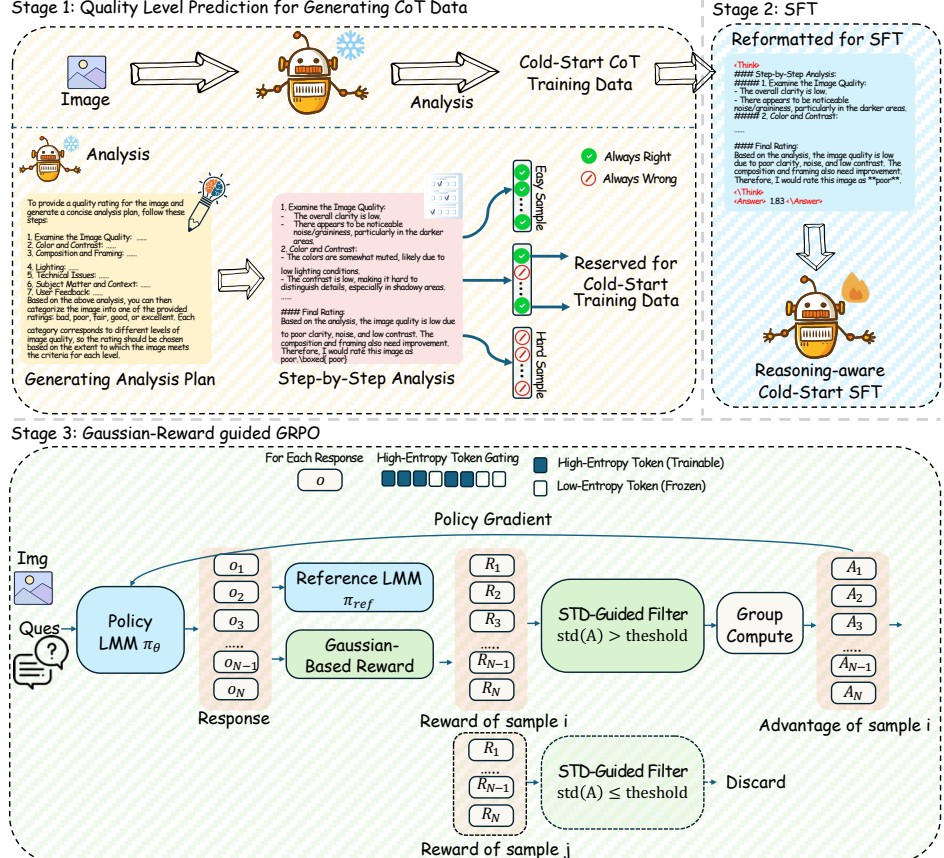

Figure 2: Overview of the OmniQuality-R Framework. The framework consists of three stages: (1) Generating Chain-of-Thought (CoT) data with quality level prediction based on structured image analysis; (2) Reasoning-aware supervised fine-tuning (SFT) using reformatted CoT samples, emphasizing hard cases; (3) Gaussian-reward guided GRPO that applies a standard deviation-based filter and high-entropy token gating to optimize policy learning, improving reasoning robustness under rule-based supervision.

refined through long-context supervised fine-tuning, enabling strong and balanced performance across diverse vision-language tasks. Specifically, they demonstrate robust capabilities on visual perception tasks (e.g., grounding, counting, OCR), visual reasoning tasks (e.g., chart understanding, table QA, math), and even GUI agent scenarios. Notably, several of these open-source models approach the performance of proprietary systems such as GPT-4o and Gemini 2.5-Flash. The second direction emphasizes enhancing visual reasoning abilities through post-training with high-quality chain-of-thought (CoT) supervised fine-tuning (SFT). For instance, Mulberry (Yao et al., 2024) employs Monte Carlo Tree Search (MCTS) with multiple MLLMs to construct long-horizon reasoning trajectories, which are then used to train reasoning-augmented models through SFT. Similarly, Mammoth-VL (Guo et al., 2024) leverages a 72B-parameter MLLM to rewrite reasoning chains for visual question answering tasks, which are subsequently used to fine-tune models.

## 2.2 MLLMs FOR IQA

Recent works have significantly advanced the fine-tuning and alignment of open-source MLLMs for IQA by introducing innovative training paradigms that balance accuracy and interpretability. Q-Instruct (Wu et al., 2024b) leveraged a novel dataset of human-authored low-level quality descriptions to instruction-tune MLLMs, markedly improving their ability to assess fine-grained distortions. In parallel, Q-Align (Wu et al., 2023a) reframed quality prediction as a classification task over discrete text-defined rating levels, aligning model outputs with human subjective rating categories for more calibrated scoring. To incorporate relative comparisons, Compare2Score (Zhu et al., 2024) trained on pairwise image comparisons and proposed a soft-anchor inference mechanism that compares test images against anchor images to infer continuous quality scores, effectively integrating diverse IQA datasets and enhancing cross-domain robustness. On the interpretability front, DepictQA (You

et al., 2024b) enabled free-form, language-based quality assessment by prompting models to generate detailed descriptions of image artifacts and comparative judgments, while Grounding-IQA (Chen et al., 2024e) further introduced spatial grounding, requiring the model to localize specific regions causing quality degradation via referring expressions, thus achieving fine-grained quality analysis. More recently, reinforcement learning has been utilized to jointly optimize scoring and reasoning capabilities. Q-Insight (Li et al., 2025c) employed a Grouped Relative Policy Optimization (GRPO) framework to refine both score regression and degradation reasoning using limited human feedback, demonstrating improved accuracy and zero-shot reasoning generalization. Similarly, VisualQuality-R1 (Wu et al., 2025) adopted a reinforcement learning-to-rank approach that shifts training from absolute scoring to relative pairwise ranking with continuous rewards, yielding superior generalization across distortions and the ability to produce human-aligned quality explanations. Moreover, Q-Ponder (Cai et al., 2025) unified score and explanation alignment in a two-stage pipeline.

## 3 METHOD

We propose OmniQuality-R, a reward model designed as a structured reasoning framework for all-encompassing image quality assessment (IQA) inspired by how expert judges evaluate images. Just as a photography judge first understands the evaluation goal, then identifies key dimensions such as composition, lighting, and artifacts to analyze individually before summarizing into an overall score, OmniQuality-R decomposes the assessment into an explicit planning stage followed by step-by-step guided reasoning. This approach encourages the model to form interpretable, dimension-aware judgments, enabling more robust and explainable reward modeling across diverse evaluation tasks, including technical quality assessment, aesthetic quality assessment, and text-image alignment. In the following, we first describe how we construct a reasoning-enhanced dataset, then present our two-stage training pipeline: supervised fine-tuning on high-quality plan-reason trajectories, and post-training with Group Relative Policy Optimization (GRPO) using a continuous reward.

### 3.1 COLD-START SFT STAGE

**Plan-then-Reason Dataset Construction** The dataset construction process follows a structured "Plan-then-Reason" methodology for three key multimodal image tasks: aesthetic rating, technical quality rating, and text-to-image alignment rating. The process begins by analyzing the given task prompt (e.g., "Please provide a technical quality rating for the image"). Based on the type of evaluation requested, the MLLM generates an analysis plan that outlines specific evaluation steps. These steps typically include examining factors such as image clarity, color and contrast, composition, lighting, and potential technical issues, among others. Once this structured plan is generated, it is paired with the original prompt and image and passed to a MLLM. The model uses this combined input to produce a detailed, step-by-step Chain-of-Thought (CoT) reasoning output. This reasoning not only clarifies the decision-making process for the final rating but also demonstrates instruction-following behavior, aligning with reasoning-aware supervision objectives.

**Rejective Sampling Finetuning** This method supports dataset creation across all three tasks by producing high-quality, traceable annotations that are used for supervised fine-tuning (SFT). Additionally, the process filters out both easy and hard examples, retaining the remaining samples for cold-start SFT training. This design mitigates the advantage vanishing issue during the subsequent reinforcement learning phase. Finally, the reasoning-enhanced MLLM $\pi_{\text{reason}}$ is fine-tuned using supervised learning on the selected CoT trajectories. Notably, during the training phase, only the original question is retained while the previously generated analysis plan is removed. This design choice is intentional and serves to bypass the need for explicit plan generation during both the reinforcement learning and inference stages. By training the model solely on the original prompt paired with the reasoning-augmented response, the approach implicitly encourages the model to internalize effective planning and reasoning structures.

### 3.2 REINFORCEMENT FINETUNING WITH GRPO

**Gaussian Reward** Previous reinforcement learning approaches for score prediction tasks, such as Q-Insight (Li et al., 2025c), typically adopt a threshold-based binary reward mechanism. These methods assign a reward of 1 if the predicted score is within a fixed margin of the ground-truth score, and 0 otherwise. While straightforward, such binary rewards are overly discrete and fail to capture the fine-grained distance between predicted and actual scores, limiting their effectiveness in continuous regression scenarios. To overcome this limitation, we propose a *Gaussian reward* formulation that produces a continuous reward signal. This design provides smoother optimization dynamics and

better aligns with the continuous nature of quality assessment tasks. Specifically, the reward is defined as $R = \exp\left(-\frac{(\hat{s}-s^*)^2}{2\sigma^2}\right)$. $\hat{s}$ denotes the model's predicted score, $s^*$ is the ground-truth score, and $\sigma$ controls the sharpness of the reward decay. This formulation ensures that predictions closer to the true value receive higher rewards, while those further away are penalized.

**Standard Deviation-Guided Sample Filtering**    To improve training efficiency and focus learning on high-quality supervision, we adopt a *standard deviation-guided filtering strategy* over sampled response groups. Let the training batch be divided into $M$ groups, each containing $K$ sampled responses with associated Gaussian rewards $\{r_j^{(i)}\}_{j=1}^K$ for group $i$. For each group, we compute its intra-group reward standard deviation $\sigma^{(i)}$, and define a filtering threshold $\tau$ to filter out over-consistent samples: $\sigma^{(i)} = \mathrm{std}(\{r_j^{(i)}\}_{j=1}^K)$, Keep group $i \iff \sigma^{(i)} > \tau, \quad \tau = 0.001$. Only groups satisfying $\sigma^{(i)} > \tau$ are retained for further training. This approach discards low-variance groups where all sampled responses yield similar rewards, which typically offer weak learning signals due to vanishing advantage. By focusing on high-variance groups, the model benefits from more informative feedback for policy gradient.

**Entropy Gating for Backward Policy Gradient**    In the later stages of training, the model typically converges on the high-level "think" structure, resulting in reduced learning signals from most tokens. To improve learning efficiency, following prior work on high-entropy minority tokens investigation (Wang et al., 2025c) and SPO (Guo et al., 2025), we adopt an entropy gating mechanism that focuses optimization on uncertain regions of the output by applying policy gradients only to high-entropy tokens. This allows the model to prioritize learning from tokens where prediction uncertainty remains high, while avoiding redundant updates on confident predictions. We modify the DAPO loss (Yu et al., 2025) (token-level loss) to apply policy gradients only on high-entropy tokens. Given a mini-batch $\mathcal{B}$ sampled from dataset $\mathcal{D}$, the loss is defined as:

$$\mathcal{J}^{\mathcal{B}}(\theta) = \mathbb{E}_{\mathcal{B}}\left[\frac{1}{\sum_{i=1}^G |o^i|}\sum_{i=1}^G\sum_{t=1}^{|o^i|}\mathbb{I}\left(H_t^i \geq \tau\right)\min\left(r_t^i(\theta)\hat{A}_t^i,\ \mathrm{clip}(r_t^i(\theta), 1-\epsilon_{\mathrm{low}}, 1+\epsilon_{\mathrm{high}})\cdot\hat{A}_t^i\right)\right] \quad (1)$$

Here, $r_t^i(\theta)$ denotes the ratio between the current policy $\pi_\theta$ and the old policy $\pi_{\mathrm{old}}$ for token $o_t^i$, i.e., $r_t^i(\theta) = \frac{\pi_\theta(o_t^i|q,o_{<t}^i)}{\pi_{\mathrm{old}}(o_t^i|q,o_{<t}^i)}$, and $\hat{A}_t^i$ is the corresponding advantage estimate. $H_t^i$ is the entropy of the predicted token distribution, and $\tau_\rho^{\mathcal{B}}$ denotes the top-$\rho$ quantile threshold computed across all token entropies within the batch. The indicator function $\mathbb{I}(H_t^i \geq \tau_\rho^{\mathcal{B}})$ ensures that only high-entropy tokens contribute to the gradient, enabling the model to focus on uncertain and informative regions.

## 3.3 REINFORCED TRAINING STRATEGY

To optimize the scoring capabilities of the multimodal model, we employ a two-stage reinforcement training strategy centered on Gaussian reward. In the initial phase, we train the policy for two epochs using only the Gaussian reward signal, which provides smooth and continuous feedback aligned with the regression nature of the task. This helps establish a strong baseline policy that can approximate human-like quality judgments. However, as training progresses, we observe a sharp decline in token-level entropy and increased convergence in predicted scores, resulting in vanishing advantage. To address this, we retain the Gaussian reward formulation but enhance the training process in the second phase by introducing two refinement mechanisms: high-entropy gating and standard deviation (STD)-guided sample filtering. The high-entropy gating mechanism modifies the GRPO objective to apply policy gradients only to high entropy tokens, encouraging the model to focus on uncertain or ambiguous output regions. Meanwhile, the STD-guided filtering computes the intra-group reward standard deviation across sampled response groups and discards low-variance groups that offer a limited gradient signal. These enhancements help maintain training efficiency and gradient informativeness during the later stages of reinforcement learning, leading to better convergence and generalization performance.

## 4 EXPERIMENT

### 4.1 EXPERIMENT SETUPS

**Training Datasets.**    We first perform SFT on a CoT dataset comprising 41,183 examples, constructed through rejective sampling from three major sources: AVA Murray et al. (2012), KonIQ Hosu

et al. (2020), and EvalMuse Han et al. (2024). This dataset spans a total of 15,206 unique images—4,916 from AVA, 4,889 from KonIQ, and 5,401 from EvalMuse—and includes 3,840 samples from AVA, 15,314 from KonIQ, and 12,029 from EvalMuse. Subsequently, Qwen2.5-VL-7B Bai et al. (2025) is fine-tuned through GRPO on three reference-scored datasets: 10K samples from AVA, 10K from EvalMuse, and 7K from KonIQ.

**Evaluation Datasets.**    We evaluate our model on a wide range of datasets across three task categories: i)**Technical Quality Assessment:** Real-scene datasets include KonIQ (Hosu et al., 2020) (excluding training images), SPAQ (Fang et al., 2020), and LIVEC (Ghadiyaram & Bovik, 2015). Synthetic distortion datasets include KADID-10k (Lin et al., 2019) and PIPAL. ii)**Aesthetic Quality Assessment:** We evaluate on AVA (Murray et al., 2012) (excluding training images) and TAD66K (He et al., 2022). iii) **Text-Image Alignment Evaluation:** Evaluation is conducted on EvalMuse (Han et al., 2024) (held-out subset), EvalMise (Wang et al., 2025a), T2I-CompBench (Huang et al., 2023),

**Test-Time Text-Image (T2I) Optimization.**    To further verify the performance of our OmniQuality-R for the text-image (T2I) generation task, we apply OmniQuality-R into the test-time guided optimization strategy for T2I generation. Specifically, we evaluate its effectiveness on the Geneval Benchmark (Ghosh et al., 2023), an object-centric benchmark that evaluates compositional properties such as position, count, and color. We test the lightweight T2I model SANA-1.0-1.6B (Xie et al., 2024) under this setting.

**Implementation Details.**    We initialize the model with Qwen2.5-VL-7B and conduct supervised fine-tuning (SFT) on the synthesis CoT dataset. For the second-stage reinforcement learning, we adopt a batch size of 64 and train for 4 epochs (2 epoches for Gaussian Reward only, and 2 epochs for the STD-filter and Entropy Gating strategy). All experiments are conducted using 4 NVIDIA A100 GPUs with 80GB memory. For GRPO, we use $n = 16$ sampled responses per query, and set the hyperparameters as $\beta = 0.04$, $\epsilon = 0.2$, and the Gaussian reward decay parameter $\sigma = 0.8$.

## 4.2   RESULT ANALYSIS

**Technical Quality Assessment Performance.**    We categorize the compared methods into three groups: handcrafted, deep-learning-based, and MLLM-based approaches. Handcrafted methods such as NIQE (Mittal et al., 2012b) and BRISQUE (Mittal et al., 2012a) rely on manually designed features and do not involve any learning process. Deep-learning-based methods (trained on KonIQ (Hosu et al., 2020)), including NIMA (Talebi & Milanfar, 2018), MUSIQ (Ke et al., 2021), CLIP-IQA++ (Wang et al., 2023), and ManIQA (Yang et al., 2022). The MLLM-based category consists of recently proposed MLLMs like Q-Align (Wu et al., 2023a), DeQA (You et al., 2025), Qwen-SFT (Bai et al., 2025), Q-Insight (Li et al., 2025c). These MLLMs follows the same joint finetuning setting with our proposed OmniQuality-R. In terms of experimental setting, all models are evaluated across six benchmark datasets: KonIQ, SPAQ, LiveW, KADID, PIPAL, and AGIQA3k, with both PLCC and SRCC used as evaluation metrics. Despite being trained jointly on three heterogeneous tasks, our method achieves consistently strong performance across all datasets, as shown in Table 1. Notably, our model reaches comparable or superior average scores to the best-performing baselines, including those trained on multiple datasets.

Table 1: PLCC / SRCC comparison on the technical quality assessment tasks with SOTA methods.

| Category | Methods | KonIQ | SPAQ | LiveW | KADID | PIPAL | AGIQA3k | AVG. |
|---|---|---|---|---|---|---|---|---|
| Training-Free *Handcrafted* | NIQE (Mittal et al., 2012b) | 0.533/0.530 | 0.679/0.664 | 0.493/0.449 | 0.468/0.405 | 0.195/0.161 | 0.560/0.533 | 0.488/0.457 |
| | BRISQUE (Mittal et al., 2012a) | 0.225/0.226 | 0.490/0.406 | 0.361/0.313 | 0.429/0.356 | 0.267/0.232 | 0.541/0.497 | 0.385/0.338 |
| | ***Test Setting*** | *In-domain* | *Out-domain* | *Out-domain* | *Out-domain* | *Out-domain* | *Out-domain* | |
| Finetune on KonIQ *Deep Learning based method* | NIMA (Talebi & Milanfar, 2018) | 0.896/0.859 | 0.838/0.856 | 0.814/0.771 | 0.532/0.535 | 0.390/0.399 | 0.715/0.654 | 0.697/0.679 |
| | MUSIQ (Ke et al., 2021) | 0.924/0.929 | 0.868/0.863 | 0.789/0.830 | 0.575/0.556 | 0.431/0.430 | 0.722/0.630 | 0.718/0.706 |
| | CLIP-IQA+ (Wang et al., 2023) | 0.909/0.895 | 0.866/0.864 | 0.832/0.805 | 0.653/0.642 | 0.427/0.419 | 0.736/0.685 | 0.737/0.718 |
| | ManIQA (Yang et al., 2022) | 0.849/0.834 | 0.768/0.758 | 0.849/0.832 | 0.849/0.465 | 0.457/0.452 | 0.723/0.636 | 0.691/0.718 |
| | ***Test Setting*** | *In-domain* | *Out-domain* | *Out-domain* | *Out-domain* | *Out-domain* | *Out-domain* | |
| Jointly Finetune *MLLM-based method* | Q-Align† (Wu et al., 2023a) | 0.936/0.934 | 0.884/0.882 | 0.869/0.860 | 0.674/0.717 | 0.443/0.420 | 0.832/0.776 | 0.774/0.764 |
| | DeQA† (You et al., 2025) | 0.928/0.908 | 0.863/0.856 | 0.847/0.866 | 0.677/0.681 | 0.411/0.390 | **0.864/0.806** | 0.763/0.751 |
| | Qwen-SFT† (Bai et al., 2025) | 0.850/0.823 | 0.870/0.862 | 0.792/0.784 | 0.643/0.659 | 0.423/0.404 | 0.793/0.658 | 0.729/0.698 |
| | Q-Insight† (Li et al., 2025c) | 0.899/0.882 | 0.899/0.897 | 0.838/0.808 | 0.627/0.666 | 0.478/**0.472** | 0.817/0.775 | 0.759/0.750 |
| | Ours | **0.941/0.927** | **0.900/0.900** | **0.880/0.857** | **0.745/0.741** | **0.481**/0.456 | 0.823/0.758 | **0.795/0.773** |

**Text-Image Alignment Quality Assessment Performance.**    Table 2 summarizes comprehensive experimental results under two evaluation settings for text-image alignment tasks. In the *Full-finetune* category, we compare several state-of-the-art methods specifically optimized for text-image alignment.

Table 2: PLCC / SRCC comparison on the text-image alignment assessment tasks.

| Category | Methods | EvalMuse-40K | EvalMi-50K | GenAI-Bench | CompBench | AVG. |
|---|---|---|---|---|---|---|
| | *Test Setting* | *Large-scale Pretraining, Out-domain testing* | | | | |
| | CLIPScore (Hessel et al., 2021) | 0.299/0.293 | 0.307/0.260 | 0.203/0.168 | 0.194/0.204 | 0.251/0.231 |
| | BLIPscore (Li et al., 2022) | 0.335/0.358 | 0.347/0.290 | 0.298/0.273 | 0.394/0.397 | 0.344/0.330 |
| | ImageReward (Xu et al., 2023) | 0.465/0.458 | 0.552/0.499 | 0.379/0.340 | 0.431/0.437 | 0.457/0.434 |
| | PickScore (Kirstain et al., 2023) | 0.440/0.433 | 0.469/0.461 | 0.363/0.354 | 0.095/0.111 | 0.342/0.340 |
| Full-Finetune | HPSv2 (Wu et al., 2023b) | 0.366/0.374 | 0.533/0.553 | 0.169/0.137 | 0.276/0.284 | 0.336/0.337 |
| | VQAScore (Li et al., 2024a) | 0.488/0.484 | 0.606/0.612 | 0.518/0.553 | 0.532/0.583 | 0.536/0.558 |
| | UnifiedReward-Llavaov (Wang et al., 2025d) | 0.710/0.722 | 0.661/0.686 | 0.606/0.621 | 0.470/0.508 | 0.612/0.634 |
| | UnifiedReward-Qwen (Wang et al., 2025d) | 0.747/0.756 | 0.736/0.717 | 0.631/0.635 | 0.627/0.648 | 0.685/0.689 |
| | *Test Setting* | *In-domain* | *Out-domain* | *Out-domain* | *Out-domain* | |
| | FGA-BLIP2 (Han et al., 2024) | 0.772/0.772 | 0.692/0.675 | 0.568/0.564 | **0.623**/0.600 | 0.664/0.653 |
| | LMM4LMM (Wang et al., 2025a) | **0.796/0.785** | 0.693/0.676 | 0.636/0.652 | 0.502/0.509 | 0.657/0.656 |
| | *Test Setting* | *In-domain* | *Out-domain* | *Out-domain* | *Out-domain* | |
| | Q-Align (Wu et al., 2023a) | 0.755/0.742 | 0.680/0.694 | 0.579/0.572 | 0.565/0.546 | 0.645/0.639 |
| | QwenSFT (Bai et al., 2025) | 0.734/0.711 | 0.718/0.683 | 0.643/0.653 | 0.566/0.598 | 0.665/0.661 |
| Few-shot Joint Finetune | DEQA (You et al., 2025) | 0.520/0.520 | 0.462/0.431 | 0.222/0.219 | 0.326/0.325 | 0.383/0.374 |
| | Q-Insight (Li et al., 2025c) | 0.647/0.688 | 0.660/0.714 | 0.641/0.672 | 0.554/0.611 | 0.626/0.671 |
| | **Ours** | 0.764/0.775 | **0.743/0.734** | **0.674/0.679** | 0.617/**0.635** | **0.700/0.706** |

Among these, FGA-BLIP2 (Han et al., 2024) and LMM4LMM (Wang et al., 2025a) are trained on the full dataset of EvalMuse-40k (Han et al., 2024), while other approaches rely on extensive large-scale pretraining (Hessel et al., 2021; Li et al., 2023; Xu et al., 2023; Kirstain et al., 2023; Wu et al., 2023b). In contrast, methods under the *Few-shot finetune* category, including ours, are jointly trained on limited few-shot datasets, as detailed in Section 4.1. Here, EvalMuse-40K serves as the in-domain dataset, whereas EvalMi-50K, GenAI-Bench, and CompBench constitute out-of-domain benchmarks. As illustrated in Table 2, our method, despite being trained with significantly fewer data samples, achieves superior performance compared to most fully fine-tuned models on the EvalMuse-40K dataset. Additionally, our approach demonstrates excellent domain generalization capabilities, outperforming even specialized full-finetuned models and achieving state-of-the-art performance across most out-of-domain benchmarks.

**Aesthetic Quality Assessment Performance.** Table 3 presents the evaluation of our approach on aesthetic quality assessment tasks under two distinct experimental settings: *Full-Finetune* and *Fewshot-Finetune*. As shown in Table 3, our method achieves state-of-the-art performance compared to all other methods in both in-domain and out-of-domain scenarios, clearly demonstrating the effectiveness and robust generalization capability.

Table 3: PLCC / SRCC comparison on the aesthetic quality assessment tasks.

| Category | Methods | AVA | TAD66k | AVG. |
|---|---|---|---|---|
| | *Test Setting* | *In-domain* | *Out-domain* | |
| Full-Finetune | MUSIQ (Ke et al., 2021) | 0.738/0.726 | 0.216/0.228 | 0.477/0.477 |
| | VILA (Ke et al., 2023) | 0.664/0.658 | 0.372/0.350 | 0.518/0.504 |
| | UNIAA (Zhou et al., 2024) | 0.704/0.713 | 0.425/0.411 | 0.565/0.562 |
| | *Test Setting* | *In-domain* | *Out-domain* | |
| | Q-Align[†] (Bai et al., 2025) | 0.747/0.729 | 0.387/0.401 | 0.567/0.565 |
| | Qwen-SFT[†] (Bai et al., 2025) | 0.650/0.629 | 0.403/0.386 | 0.526/0.507 |
| Few-shot Joint Finetune | DeQA[†] (You et al., 2025) | 0.751/0.749 | 0.433/0.409 | 0.592/0.579 |
| | Q-Insight[†] (Li et al., 2025c) | 0.699/0.699 | 0.443/0.416 | 0.571/0.558 |
| | ours | **0.766/0.763** | **0.461/0.430** | **0.612/0.597** |

Table 4: Performance comparison across Stage 1 and Stage 2 reinforcement training. Stage 2 training is initialized from the corresponding Stage 1 checkpoint.

| Stage | Method | In-domain | | | Out-of-domain | | |
|---|---|---|---|---|---|---|---|
| | | AVA | EvalMuse-50k | KonIQ | TAD66k | EvalMi-50k | KADID |
| | GRPO+GR | **0.763 / 0.760** | **0.764 / 0.768** | 0.937 / 0.925 | **0.448/0.406** | 0.740/0.731 | 0.672/0.675 |
| Stage 1 | + Entro. | 0.762 / 0.762 | 0.742 / 0.746 | **0.938 / 0.925** | 0.437/0.397 | 0.716/0.715 | 0.645/0.665 |
| | + Entro. + STD | 0.757 / 0.759 | 0.748 / 0.748 | 0.934 / 0.920 | 0.436/0.398 | **0.738/0.742** | **0.735/0.731** |
| | GRPO+GR | 0.760 / 0.755 | 0.750 / 0.751 | 0.936 / 0.922 | 0.445/0.412 | 0.734/0.728 | 0.671/0.672 |
| Stage 2 (from GRPO+GR) | + Entro. | **0.767 / 0.765** | 0.767 / 0.771 | 0.940 / 0.927 | 0.450/0.409 | 0.739/0.735 | 0.735/0.740 |
| | + Entro. + STD | 0.766 / 0.763 | **0.764 / 0.775** | **0.941 / 0.927** | **0.461/0.430** | **0.743/0.734** | **0.745/0.741** |

Table 5: Results on the GenEval benchmark

| Generator | Overall | Single | Two | Counting | Color | Position | Attribution |
|---|---|---|---|---|---|---|---|
| SANA-1.5–4.8B (Xie et al., 2025b)[‡] | 0.76 | 0.99 | 0.95 | 0.72 | 0.82 | 0.50 | 0.54 |
| + Best-of-2048[‡] | 0.80 | 0.99 | 0.88 | 0.77 | **0.90** | 0.47 | **0.74** |
| SANA-1.0–1.6B[†] (Xie et al., 2025a) | 0.62 | 0.98 | 0.83 | 0.58 | 0.86 | 0.19 | 0.37 |
| + Best-of-20 | 0.75 | 0.99 | 0.87 | 0.73 | 0.88 | 0.54 | 0.55 |
| **+ Best-of-20 with OmniQuality-R** | 0.78 | 0.98 | **0.96** | 0.76 | 0.86 | 0.58 | 0.60 |
| **+ Best-of-20 and Reflected 20 times with OmniQuality-R** | **0.80** | **1.00** | **0.96** | **0.78** | 0.86 | **0.66** | 0.61 |

**Visual Reward-Guided Test-Time T2I Scaling.** As shown in Table 5, we report the performance of T2I test-time scaling guided by our OmniQuality-R on the GenEval (Ghosh et al., 2023) benchmark.

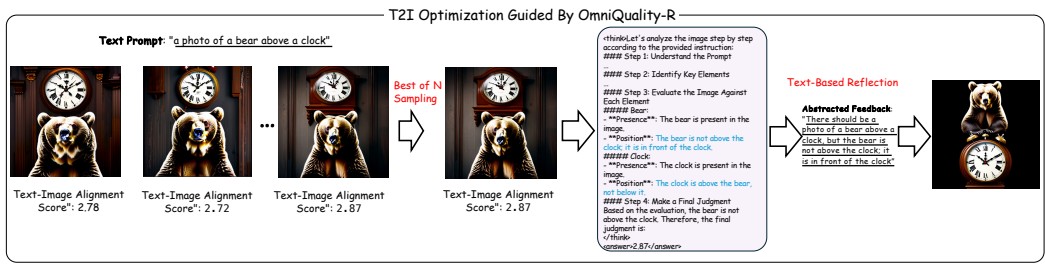

Figure 3: The T2I optimization process visualization guided by our proposed OmniQuality-R.

Table 6: Results on the main component of our OmniQuality-R framework with 2 epochs training setting for stage 1 reinforcement training.

| Method | AVA | EvalMuse-50k | KonIQ |
|---|---|---|---|
| Naive GRPO | 0.733/0.732 | 0.640/0.641 | 0.899/0.882 |
| Rej. Tuning + GRPO | 0.737/0.735 | 0.689/0.706 | 0.911/0.897 |
| Rej. Tuning + GPRO with GR | **0.763/0.760** | **0.764/0.768** | **0.937/0.925** |

Our method achieves the optimal performance on the GenEval benchmark through a two-stage enhancement strategy combining score-guided selection and text-guided reflection. Starting from the base SANA-1.0–1.6B model (Xie et al., 2024), naive Best-of-20 sampling yields a solid baseline (0.75 overall). By incorporating score prediction (OmniQuality-R) to guide the Best-of-20 sampling, we obtain a significant improvement to 0.78, outperforming the larger SANA-1.5–4.8B (0.76) despite using significantly fewer parameters and less compute. Building on this, we further apply the text-guided reflection strategy described in Reflect-DiT (Li et al., 2025b) to iteratively refine the generated outputs, achieving a new state-of-the-art score of 0.80 overall. While the larger SANA-1.5–4.8B model with Naive Best of 2048 sampling performs competitively, it is not open-sourced and requires significantly more computational resources and cost. The optimization process is shown in Fig. 3.

## 5 ABLATION STUDY

**The effectiveness of cold-start reject sampling tuning** As shown in Tab 6, cold-start reject sampling leads to consistent improvements over the naive GRPO baseline, especially on more challenging datasets like EvalMuse (0.640→0.689). These results facilitate the learning of reasoning pathway structures that reflect the underlying thought processes.

**The effectiveness of Gaussian Reward** Comparing the second and third rows of Tab 6, we observe that replacing the binary reward with a Gaussian-shaped reward leads to consistent performance gains. This suggests that continuous reward modeling provides richer learning signals for the policy.

**The effectiveness of Entropy Gating and STD-guided Filtering** Tab 4 reveals that directly applying entropy-based gating or STD-guided filtering during Stage 1 reinforcement learning leads to slight performance degradation across most benchmarks. And we found that these mechanisms can be counterproductive when applied to a model that has not yet acquired a strong quality assessment baseline—often resulting in unstable optimization or degraded performance. In contrast, when introduced in Stage 2 (after warming up the policy with Gaussian reward), their effects become significantly positive. The model benefits from stabilized reasoning pathways and gains better learning signals, resulting in consistent performance boosts.

## 6 CONCLUSION

In this work, we have presented OmniQuality-R, a reward model with a structured reasoning framework for explainable and robust image quality assessment across multiple tasks. OmniQuality-R decomposes reward modeling over key quality dimensions and leverages high-quality plan-reason trajectories for fine-grained, interpretable assessment. It combines supervised fine-tuning on structured CoT data with GRPO-based reinforcement learning for continuous score prediction, further stabilized by STD filtering and entropy gating. Extensive experiments across aesthetic, technical, and alignment evaluation tasks demonstrate that OmniQuality-R not only achieves structured reasoning across diverse settings, but also substantially improves generalization to unseen data, suggesting a scalable path toward more robust reward models. Moreover, OmniQuality-R can serve as a test-time reward model for guiding text-to-image (T2I) generation models, highlighting its versatility and effectiveness.

## ETHICS STATEMENT

Our research on Quality Assessment is purely algorithmic and all experiments were conducted on publicly available benchmark datasets. This study did not involve human or animal subjects, and we foresee no direct negative societal impacts or ethical concerns arising from our work.

## REPRODUCIBILITY STATEMENT

To ensure reproducibility, all source code for our method, along with the experimental scripts, will be made available in a public repository upon publication, The datasets used are standard public benchmarks, and our repository will include detailed instructions to replicate all reported results.

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

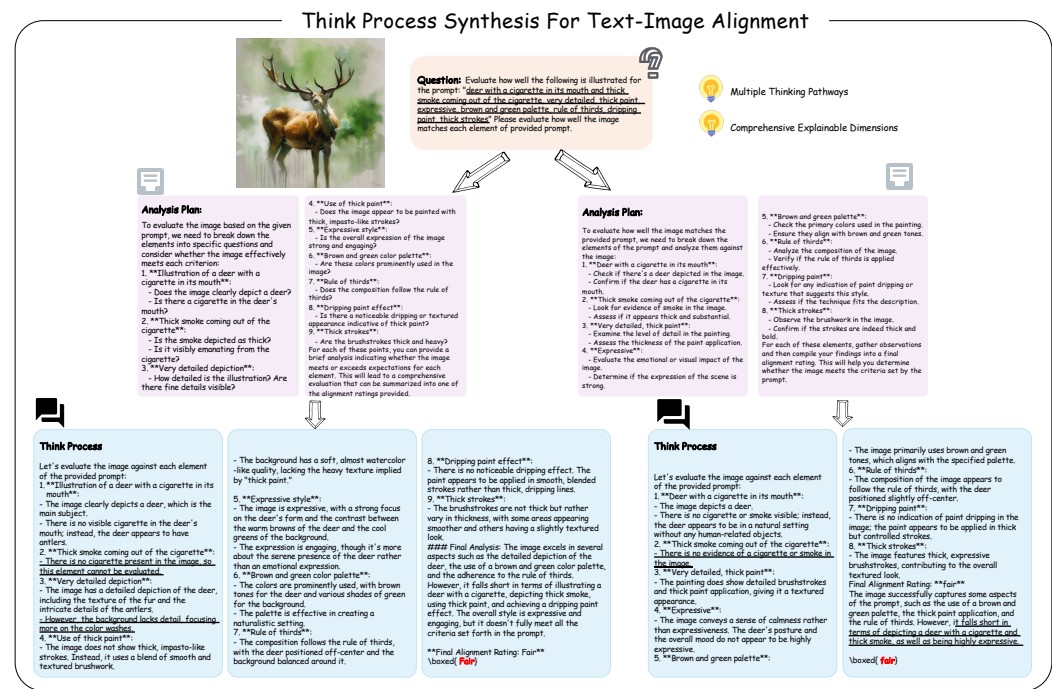

Figure 4: The visualization of think process synthesis for text-image alignment.

# A APPENDIX

## A.1 USE OF LANGUAGE MODELS FOR MANUSCRIPT EDITING

This manuscript has benefited from language refinement using a large language model (LLM). The LLM was employed exclusively for linguistic improvements, such as enhancing clarity, grammar, and style, without altering the scientific content, data interpretation, or conclusions. All substantive ideas, analyses, and findings presented herein are entirely the authors' own, and the responsibility for the final content rests solely with the authors.

## A.2 DATASET GENERATION

The dataset generation processes for the text-image alignment task, aesthetic quality assessment task, and technical quality assessment task are illustrated in Fig. 4, Fig. 5, and Fig. 6, respectively.

Each image is paired with one or more chain-of-thought (CoT) reasoning paths generated from Qwen2.5-VL-7B. The average response lengths also vary across datasets, with AVA exhibiting longer reasoning trajectories (mean: 478.04 tokens), while KonIQ and EvalMuse maintain more concise styles (mean: 346.17 and 342.39 tokens, respectively). This diversity in both image content and reasoning style promotes broader generalization and robustness in quality assessment capabilities.

## A.3 MORE EXPERIMENTS

**Token Entropy Visualization.** To better understand how high entropy gating impacts the model's reasoning behavior, we visualize the token-level entropy heatmaps before and after reinforcement training with entropy gating across all three tasks. As shown in Fig. 8, Fig. 9, and Fig. 10, after training, the model generates more high-entropy tokens, especially around key reasoning and decision points. This indicates increased uncertainty awareness and richer lexical exploration. Our entropy-gated training effectively mitigates entropy collapse and encourages more informative and diverse reasoning.

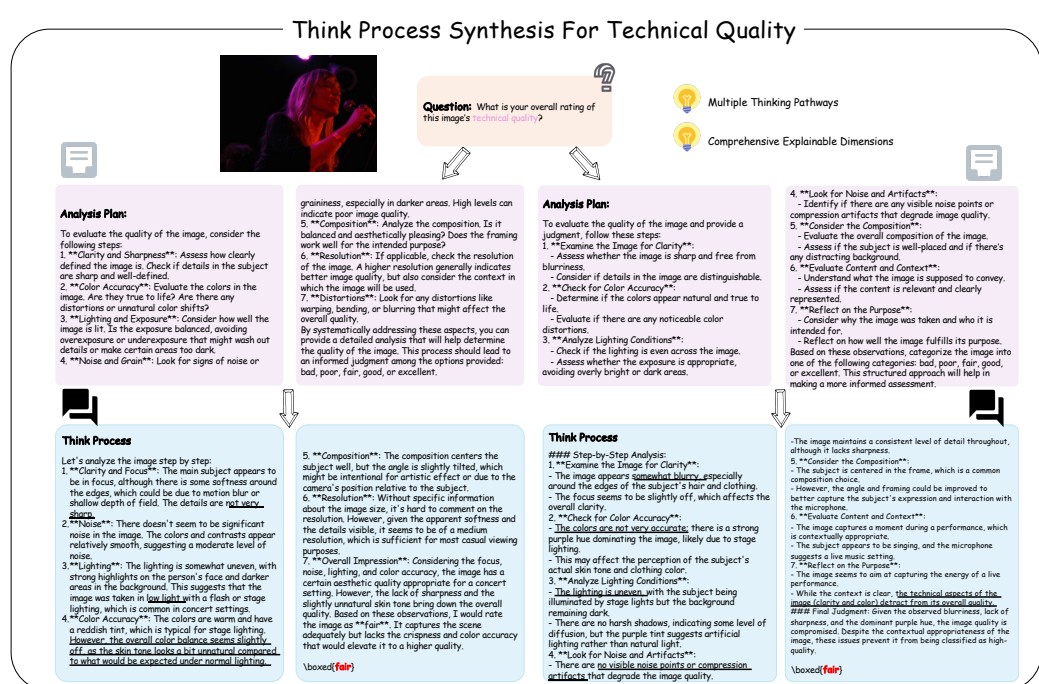

Figure 5: The visualization of think process synthesis for image technical quality assessment.

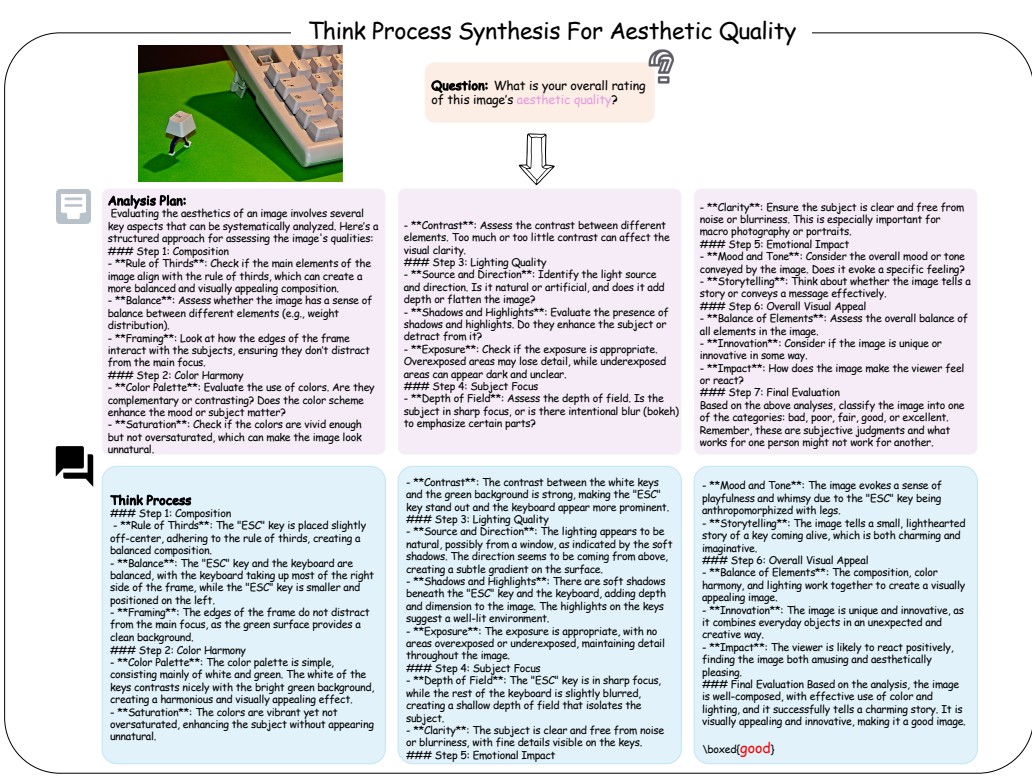

Figure 6: The visualization of think process synthesis for image aesthetic quality assessment.

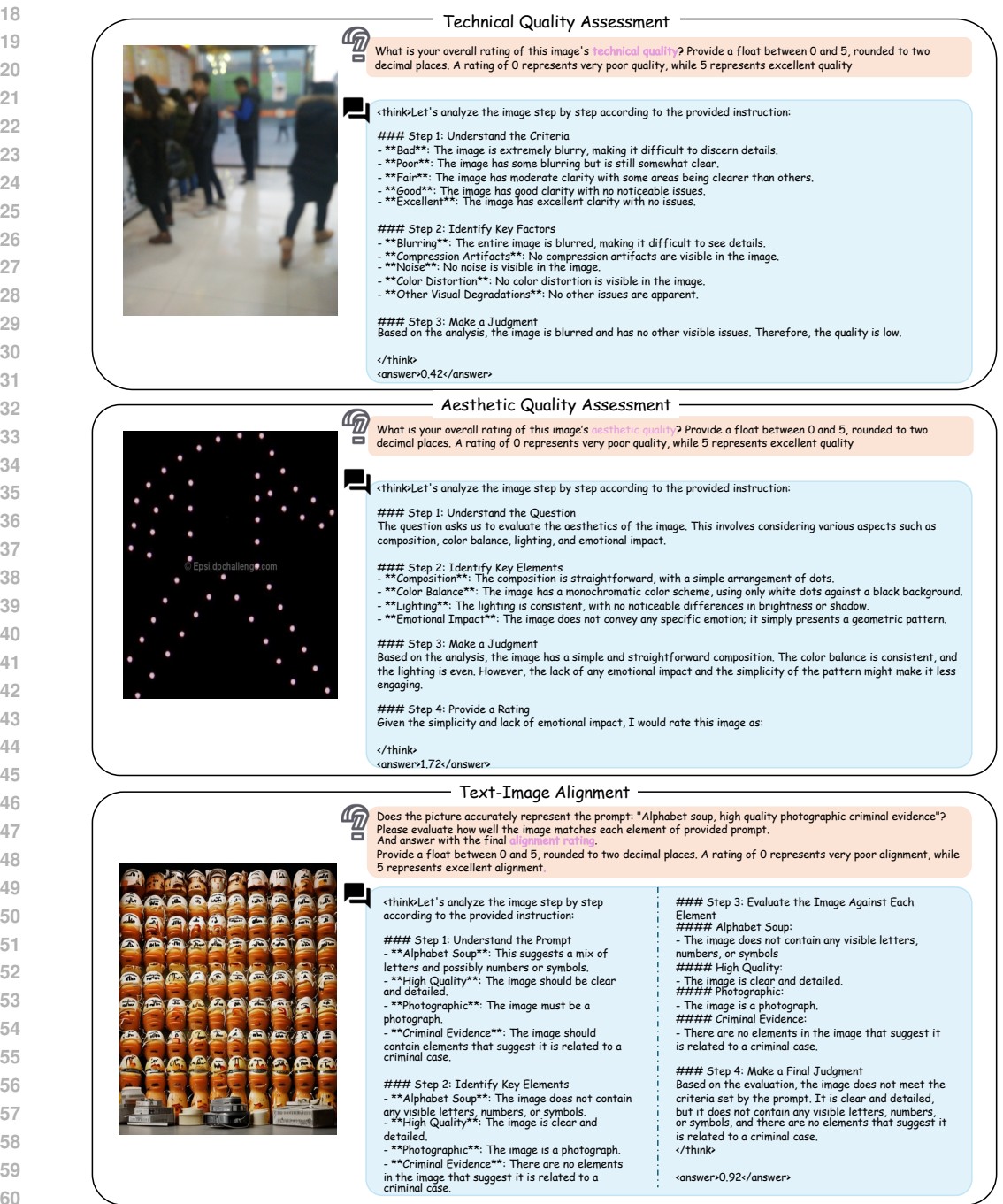

Figure 7: The visualization of our proposed OmniQuality-R on three evaluation tasks.

**Gaussian Reward Visualization.** From the Fig. 11, it can be observed that the threshold-based reward remains fixed at 1 when the error is less than 0.3, but drops abruptly to 0 once the threshold is exceeded, showing a lack of continuity. In contrast, the Gaussian-based reward decreases smoothly as the error increases, and even beyond the threshold the model still receives feedback proportional to the prediction quality. As a result, the Gaussian reward provides more informative guidance, accelerates convergence, and improves model performance on continuous-valued quality evaluation tasks.

**More application on test-time guidance.** To evaluate the effectiveness of test-time guidance in text-to-image (T2I) generation, we adopt a best-of-20 sampling strategy, where 20 images are

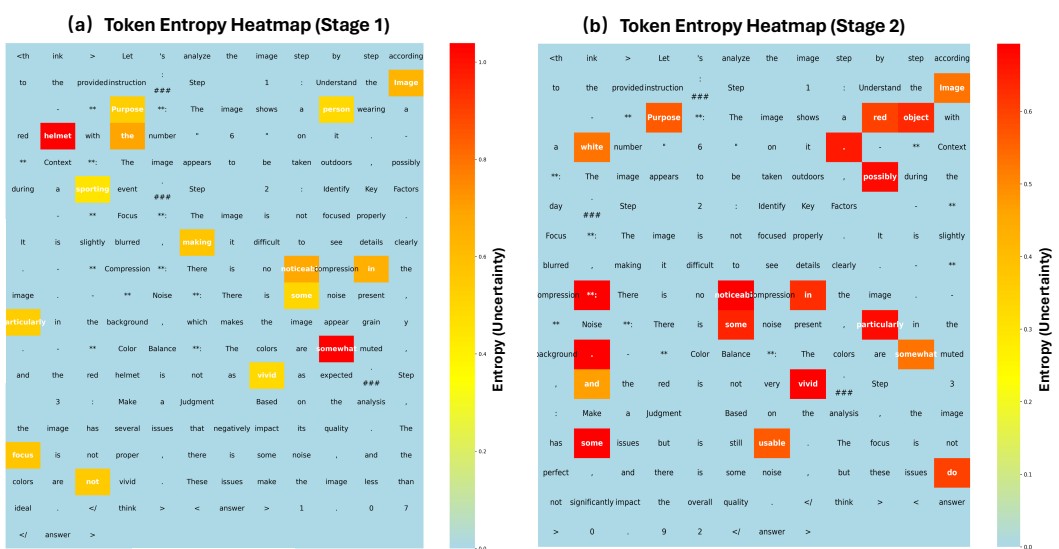

Figure 8: The entropy distribution for each word in the response of technical quality assessment.

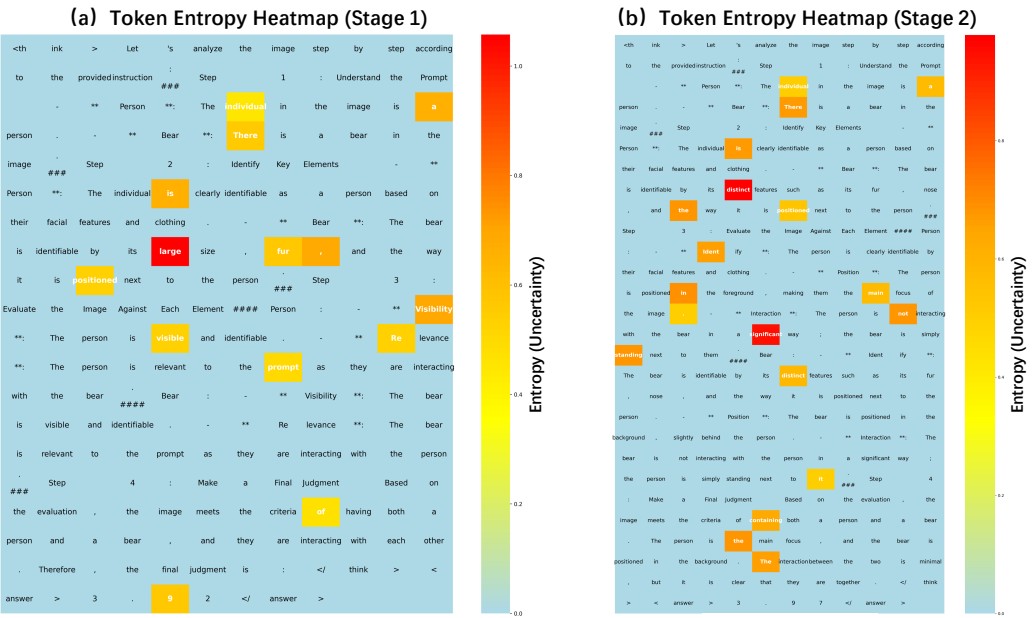

Figure 9: The entropy distribution for each word in the response of text-image alignment.

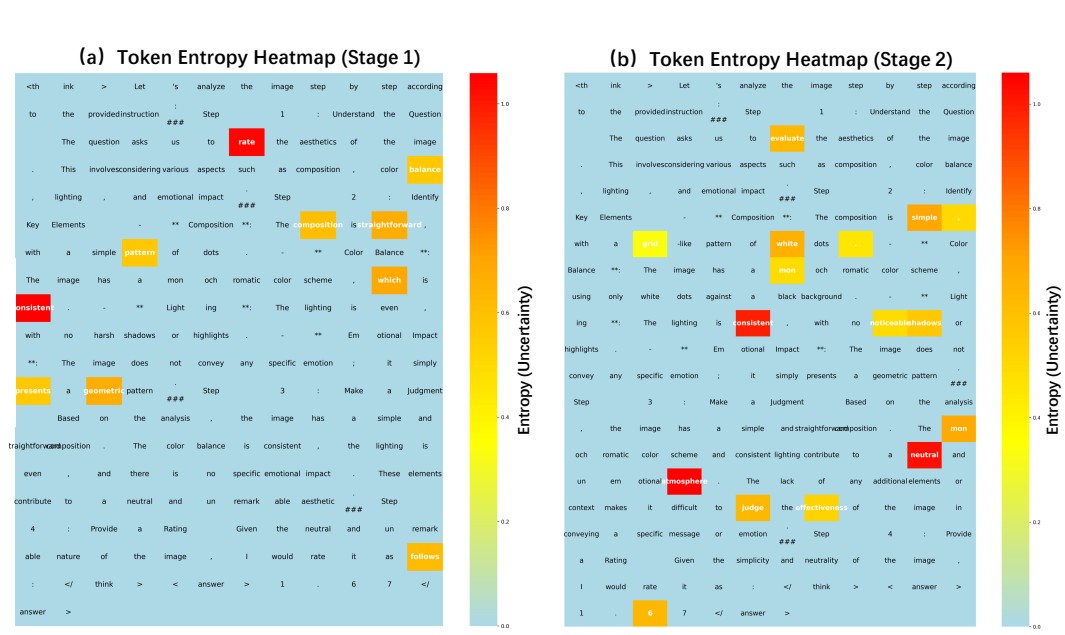

Figure 10: The entropy distribution for each word in the response of aesthetic quality assessment.

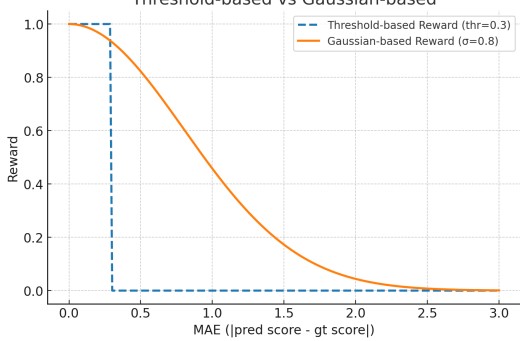

Figure 11: Comparison of threshold-based and Gaussian-based reward functions.

generated per prompt and a scoring-based selector is used to choose the final output. We compare three selection methods on SANA 1.0: (1) Q-Align select, which uses Q-Align (Wu et al., 2023a)'s aesthetic score; (2) OmniQuality-R-Aes select, which selects based on OmniQuality-R's aesthetic score alone; and (3) OmniQuality-R select, which combines both aesthetic and technical scores from OmniQuality-R. The selected images are then evaluated using three external metrics: Q-Insight (Li et al., 2025c), DEQA (You et al., 2025), and AesMMIT (Huang et al., 2024).

As shown in Table 7, the OmniQuality-R selector achieves the highest scores across all metrics, outperforming both Q-Align and the aesthetic-only variant. This indicates that considering both aesthetic and technical quality dimensions leads to more robust and perceptually preferred image selection. Notably, while aesthetic-only selection already improves over Q-Align, incorporating technical quality provides further performance gains, suggesting its complementary importance in high-quality T2I generation.

Table 7: Comparison of generated images selected by different methods (20 images per prompt). Higher scores indicate better performance. And the final score is averaged on a subset of 50 prompts.

| Selection Method | Q-Insight ($\uparrow$) | DEQA ($\uparrow$) | AesMMIT ($\uparrow$) |
|---|---|---|---|
| Q-Align select | $4.165 \pm 0.198$ | $3.897 \pm 0.461$ | $0.480 \pm 0.114$ |
| OmniQuality-R-Aes select | $4.208 \pm 0.175$ | $3.912 \pm 0.441$ | $0.513 \pm 0.116$ |
| OmniQuality-R select | $4.264 \pm 0.175$ | $4.062 \pm 0.363$ | $0.557 \pm 0.101$ |

Table 8: Results on the Gaussian reward decay parameter ablation in Stage 1 Training.

| $\sigma$ | AVA | EvalMuse-50k | KonIQ |
|---|---|---|---|
| 0.6 | 0.761/0.756 | 0.734/0.756 | 0.931/0.929 |
| 0.8 | **0.763/0.760** | **0.764/0.768** | **0.937/0.925** |
| 1 | 0.741/0.735 | 0.724/0.737 | 0.913/0.912 |

**Hyper-Parameter.** Table 8 shows that the Gaussian reward decay achieves the best results at $\sigma = 0.8$, striking a balance between over-penalization (when $\sigma$ is too small) and under-penalization (when $\sigma$ is too large).

