# OpenReview forum: "OmniQuality-R: Advancing Reward Models through All-Encompassing Quality Assessment"
_ICLR.cc/2026/Conference — ICLR 2026 Conference Withdrawn Submission_

### Official Review · Reviewer_3eej · 2025-10-27

**Soundness:** 4
**Presentation:** 3
**Contribution:** 2
**Rating:** 4
**Confidence:** 3

**Summary:**

This paper introduces a multi-task reward modeling framework OmniQuality-R for image quality assessment. It constructs a reliable chain-of-thought dataset to pre-train MLLM for reward modeling and then proposes to use continuous reward, standard deviation filtering, and entropy gating mechanisms to train the model with GRPO. It demonstrates OmniQuality-R can produce state-of-the-art performance on image quality assessment tasks.

**Strengths:**

- The reward modeling pipeline is robust and the quantitative results demonstrate the effectiveness of the pipeline. This can be of interest to the community who work on reward modeling and generative models.
- The sufficient detail provided in the text enables reproduction of the paper feasible.
- The paper is written clearly and easy to follow.

**Weaknesses:**

- While the proposed pipeline is clear and robust, the main question arises from the technical contribution of the proposed method.
The main pipeline is quite similar to the one proposed in "Q-Ponder: A Unified Training Pipeline for
Reasoning-based Visual Quality Assessment" (Can et al. 2025), including the way it treats the reward with a continuous function and creating CoT dataset that target multiple aspects for cold-start initialization, even though the exact target task is different.
- The related work section could highlight the difference between the current work and related works.

Due to the above point, the reviewer currently feels the work is incremental

**Questions:**

As the reviewer agrees with the technical robustness of the pipeline, the main point that could be addressed in the rebuttal is regarding the weaknesses section above.

---

### Official Review · Reviewer_csT7 · 2025-10-29

**Soundness:** 2
**Presentation:** 2
**Contribution:** 2
**Rating:** 2
**Confidence:** 4

**Summary:**

This paper propose OmniQuality-R, a unified framework for reward modeling in image quality assessment. It try to combine multiple aspects like technical quality, aesthetic appeal and text-image alignment into one model. Inspired by how human evaluate images, it use structured reasoning with cot and then apply rl with GRPO. They introduce Gaussian reward instead of binary, and add STD filtering and entropy gating to make training more stable. The model is evaluate on three tasks and claim to improve robustness, explainability and can even guide T2I generation at test time without retraining.

**Strengths:**

- The idea of all-encompassing quality assessment is timely, with rise of AIGC and need for better reward models in generative AI. Current methods are limited to single task, this one integrate them nicely.

- Two-stage training process seem innovative: first cold-start SFT with rejective sampling to build good CoT data, then RL with Gaussian reward for continuous scoring. This could help in other regression tasks beyond IQA.

- Mechanisms like STD-guided filtering and entropy gating address common issue in RL like vanishing advantage and low entropy, which make training unstable. Good to see ablation or discussion on this.

- Evaluation cover diverse tasks: aesthetic (AVA), technical (KonIQ?), alignment. And it show application in guiding T2I models, which is practical.

- Figure 1 and 2 are clear and help understand the framework.

**Weaknesses:**

- Dataset construction rely on rejective sampling, but how to ensure the "hard" and "easy" sample filtering is fair? Might bias the model.

- It would be better if include more ablation on component like without Gaussian reward or without entropy gating. Also, cross-domain generalization not fully address, e.g., on real-world UGC vs synthetic distortion.

- Writing have some repetition, like "all-encompassing" use too much, and some sentence long and hard to follow.
Not discuss computational cost – MLLM fine-tuning with RL is expensive, how scalable it is?

**Questions:**

- In Gaussian reward, how to choose sigma? Is it tune per task or fixed and how?

- For test-time guidance in T2I, can you give example of how reward is use – like in sampling or optimization loop?

- Why focus only on three tasks? Could it extend to video quality or 3D?

- In rejection sampling, what MLLM is use to generate plan and reason – is it same base model?

---

### Official Review · Reviewer_PjR5 · 2025-10-31

**Soundness:** 3
**Presentation:** 3
**Contribution:** 3
**Rating:** 4
**Confidence:** 4

**Summary:**

This paper introduces OmniQuality-R, a unified reward model for image quality assessment covering technical quality, aesthetics, and text-image alignment. It uses a structured plan-and-reason approach with chain-of-thought data and a two-stage training process combining supervised fine-tuning and reinforcement learning with a Gaussian reward. Stabilized by entropy gating and standard deviation filtering, OmniQuality-R achieves strong generalization, interpretability, and effectively guides text-to-image generation.

**Strengths:**

The advantages of this paper are as follows:
1.OmniQuality-R separates the planning and reasoning stages, enabling structured and interpretable evaluations across multiple image quality dimensions.
2.The incorporation of standard deviation (STD) guided filtering and entropy gating effectively improves the Group Relative Policy Optimization (GRPO) by reducing variance and enhancing generalization.
3.Unlike previous works that use binary rewards, this paper adopts a continuous Gaussian reward, which is practical and effective for score regression tasks.
4.The proposed method achieves state-of-the-art performance on multiple benchmarks and demonstrates strong out-of-domain generalization capabilities.

**Weaknesses:**

1.The overall training pipeline is largely similar to the previous UnifiedReward-Think approach, and the interpretability and stability of step-by-step Chain-of-Thought (CoT) rewards have already been discussed in UnifiedReward-Think. Moreover, the claim of being "all-encompassing" is not well substantiated in this paper. It is recommended that the authors clarify why they chose these particular dimensions and how these choices contribute to Image Quality Assessment
2.Regarding Test-Time Text-Image (T2I) Optimization, the SANA-1.0-1.6B model used is relatively small and not mainstream. The evaluation focuses only on semantic improvements, and the visualization is limited to positional optimization.  Efficiency in aesthetic and image quality optimization need be explored
3.The STD filtering and Entropy Gating discard some rollout samples. Poor hyperparameter choices might lead to the need for more rollouts. It is suggested that the authors provide experimental comparisons of GRPO with different numbers of function evaluations (NFE) to illustrate this effect.

**Questions:**

Please refer to the Weaknesses section above for details.

---

### Official Review · Reviewer_FyQ2 · 2025-10-31

**Soundness:** 2
**Presentation:** 2
**Contribution:** 2
**Rating:** 4
**Confidence:** 3

**Summary:**

This paper introduces OmniQuality-R, a unified reward model designed for "all-encompassing" IQA. The authors identify that current IQA models are typically specialized for a single task (e.g., technical quality, aesthetics, or text-image alignment), lack interpretability, and generalize poorly.

**Strengths:**

Pros:
1. The paper has an ambitious goal of unifying three distinct IQA tasks into a single, cohesive framework. This "all-encompassing" perspective is a step beyond narrow, single-task models and aligns better with the multifaceted nature of human quality perception.
2. The "plan-then-reason" structure and use of CoT data directly tackle the black-box nature of many IQA models. The model doesn't just output a score, it provides a step-by-step rationale.
3. The authors evaluate their model on an impressive number of datasets covering all three tasks, including both in-domain and out-of-domain scenarios. The consistent top-tier performance across these benchmarks strongly validates their approach.

**Weaknesses:**

Cons:
1. The quality of the entire framework hinges on the initial CoT dataset generated by the Qwen2.5-VL-7B "teacher" model. The paper does not discuss the potential limitations or biases inherited from this teacher. A flawed teacher could propagate systematic errors or a particular "style" of reasoning into OmniQuality-R.
2. The technical contribution is not very clear since the training components have already existed. The multi-stage pipeline (CoT generation, rejective sampling, SFT, and a customized GRPO training) is highly complex but is not quite novel.
3. The paper excels at showcasing positive results but would be strengthened by an analysis of its failure modes. Which types of images or prompts does it struggle with? When does its reasoning break down? This would provide a more balanced view of the model's capabilities and limitations.
4. Aesthetic assessment is inherently subjective and culturally dependent. The paper could benefit from a discussion on this limitation and how the framework might be adapted to handle diverse or personalized aesthetic preferences.
5. More Dataset Generation details are desired. The "Plan-then-Reason" dataset construction is a critical first step. Was there any human verification of the auto-generated CoT trajectories, or was the process fully automated? How sensitive is the final performance to the choice and quality of this initial teacher MLLM?
6. Could you elaborate on the specific criteria used to classify examples as "easy" or "hard" for the rejective sampling finetuning? Is this based on prediction error, response length, or another heuristic?
7. In Table 7, the best-performing "OmniQuality-R select" method combines both aesthetic and technical scores. How were these two scores combined? Was it a simple sum, a weighted average, or a more complex function? How was this combination strategy determined?
8. The analysis plan is generated in Stage 1 but removed for SFT to encourage the model to "internalize" planning. Have you explored an alternative where the model explicitly generates a plan first during inference? Could this potentially improve robustness on out-of-distribution or particularly complex assessment tasks?

**Questions:**

See Weaknesses

---

### Note · Authors · 2025-11-26

**Comment:**

thanks for the reviewers and AC's efforts

**Withdrawal Confirmation:**

I have read and agree with the venue's withdrawal policy on behalf of myself and my co-authors.